# Weakly-supervised Knowledge Graph Alignment with Adversarial Learning

## Abstract

This paper studies aligning knowledge graphs from different sources or languages. Most existing methods train supervised methods for the alignment, which usually require a large number of aligned knowledge triplets. However, such a large number of aligned knowledge triplets may not be available or are expensive to obtain in many domains. Therefore, in this paper we propose to study aligning knowledge graphs in fully-unsupervised or weakly-supervised fashion, i.e., without or with only a few aligned triplets. We propose an unsupervised framework to align the entity and relation embddings of different knowledge graphs with an adversarial learning framework. Moreover, a regularization term which maximizes the mutual information between the embeddings of different knowledge graphs is used to mitigate the problem of mode collapse when learning the alignment functions. Such a framework can be further seamlessly integrated with existing supervised methods by utilizing a limited number of aligned triples as guidance. Experimental results on multiple datasets prove the effectiveness of our proposed approach in both the unsupervised and the weakly-supervised settings.

## 1 Introduction

Knowledge graphs represent a collection of knowledge facts and are quite popular in the real world. Each fact is represented as a triplet $(\mathbf{h}, \mathbf{r}, \mathbf{t})$, meaning that the head entity $\mathbf{h}$ has the relation $\mathbf{r}$ with the tail entity $\mathbf{t}$. Examples of real-world knowledge graphs include instances which contain knowledge facts from general domain (e.g., Freebase [1], WordNet [2]) or facts from specific domains such as biomedical ontology (e.g., UMLS [3]). Knowledge graphs are critical to a variety of applications such as question answering (Bordes et al., 2014) and semantic search (Guha et al., 2003). Research on knowledge graphs is attracting growing interest recently in both academia and industry communities.

In practice, each knowledge graph is usually constructed from a single source or language, the coverage of which is limited. To enlarge the coverage and construct more unified knowledge graphs, a natural idea is to integrate multiple knowledge graphs from different sources or languages (Arens et al., 1993). However, different knowledge graphs use distinct symbol systems to represent entities and relations, which are not compatible. Therefore, it is critical to align the entities and relations across different knowledge graphs (a.k.a., knowledge graph alignment) before integrating them.

Recently, many methods have been proposed to align entities and relations from a source knowledge graph to a target knowledge graph (Zhu et al., 2017a; Chen et al., 2017a;b; Sun et al., 2018a). These methods first represent the entities and relations in low-dimensional spaces and then learn mapping functions to align the entities and relations from the source knowledge graph to the target one. Though these methods are proven quite effective, they rely on a large number of aligned triplets for training supervised alignment models, and such aligned triplets may not be available or can be expensive to obtain. As a result, the performance of these methods will be comprised. Therefore, it would be desirable to design an unsupervised or weakly-supervised approach for knowledge graph alignment, which requires a few or even without aligned triplets.

---

[1] https://developers.google.com/freebase/
[2] https://wordnet.princeton.edu/
[3] https://www.nlm.nih.gov/research/umls/

In this paper, we propose an unsupervised approach to knowledge graph alignment with adversarial training (Goodfellow et al., 2014). Our proposed approach first represents the entities and relations in low-dimensional spaces with existing knowledge graph embedding methods (e.g., TransE (Bordes et al., 2013)) and then learns alignment functions, i.e., $p_e(\mathbf{e}_t|\mathbf{e}_s)$ and $p_r(\mathbf{r}_t|\mathbf{r}_s)$, to map the entities and relations ($\mathbf{e}_s$ and $\mathbf{r}_s$) from the source knowledge graph to those ($\mathbf{e}_t$ and $\mathbf{r}_t$) in the target graph. Intuitively, an ideal alignment function is able to map all triples in the source graph to valid ones in the target graph. Therefore, we train a triplet discriminator to distinguish between the real triplets in the target graph and those aligned ones from the source graph. Such a discriminator measures the plausibility of a triplet in the target graph and provides a reward function for optimizing the alignment functions, which are optimized to fool the discriminator. The above process naturally forms an adversarial training procedure. By alternatively optimizing the alignment functions and the discriminator, the whole process can constantly enhance the alignment functions.

Despite the effectiveness of adversarial learning in many scenarios, one big problem it may suffer from is the mode collapse (Salimans et al., 2016). Specifically, in our case, it means that many entities in the source knowledge graph are aligned to only a few entities in the target knowledge graph. We propose to mitigate this problem by maximizing the mutual information between the entities in the source graph and those aligned entities, which can be effectively and effectively optimized with some recent techniques on mutual information neural estimation (Belghazi et al., 2018). We further prove that by maximizing the mutual information, different source-graph entities are encouraged to be aligned to different target-graph entities, which mitigates the mode collapse.

The whole framework can also be seamlessly integrated with existing supervised methods, in which we can use a few aligned entities or relations as guidance, yielding a weakly-supervised approach. Our approach can be effectively optimized with stochastic gradient descent, where the gradient for the alignment functions is calculated by the REINFORCE algorithm (Williams, 1992). We conduct extensive experiments on several datasets. Experimental results prove the effectiveness of our proposed approach in both the weakly-supervised and unsupervised settings.

## 2 RELATED WORK

Our work is related to knowledge graph embedding, which represents entities and relations as low-dimensional vectors (a.k.a., embedding). A variety of approaches have been proposed (Bordes et al., 2013; Wang et al., 2014; Yang et al., 2014; Sun et al., 2018b), which can effectively preserve the similarities of entities and relations into the learned embeddings. We treat these techniques as tools to learn entity and relation embeddings, which serve as features for knowledge graph alignment.

In literature, there are also some studies focusing on knowledge graph alignment. Most of them perform alignment by considering contextual features of entities and relations, such as their names (Lacoste-Julien et al., 2013) or text descriptions (Chen et al., 2018; Wang et al., 2012; 2013). However, such contextual features are not always available, and therefore these methods cannot generalize to most knowledge graphs. In this paper, we consider the most general case, in which only the triplets in knowledge graphs are used for alignment. The studies most related to ours are Zhu et al. (2017a), Chen et al. (2017a) and Sun et al. (2018a). Similar to our approach, they treat the entity and relation embeddings as features, and jointly train an alignment model. However, they totally rely on the labeled data (e.g., aligned entities) to train the alignment model, whereas our approach incorporates additional signals by using adversarial training, and therefore achieves better results in the weakly-supervised and unsupervised settings.

More broadly, our work belongs to the family of domain alignment, which aims at mapping data from one domain to the other domain. With the success of generative adversarial networks (Goodfellow et al., 2014), many researchers have been bringing the idea to domain alignment, getting impressive results in many applications, such as image-to-image translation (Zhu et al., 2017b;c), word-to-word translation (Conneau et al., 2017) and text style transfer (Shen et al., 2017). These studies typically train a domain discriminator to distinguish between data points from different domains, and then the alignment function is optimized by fooling the discriminator. Our approach shares similar idea, but is designed with some specific intuitions in knowledge graphs.

Besides, our work is also related to recent studies on neural mutual information estimation (Belghazi et al., 2018), which estimate the mutual information of two distributions by using neural networks.

Such a technique has been utilized in many applications, including image classification (Hjelm et al., 2018) and unsupervised node representation learning (Veličković et al., 2018). All these studies use the technique to improve representation learning (e.g., image representation, node representation). In contrast, our approach uses the technique to avoid mode collapse in adversarial learning.

Finally, Cai & Wang (2018) propose a method called KBGAN recently, which uses adversarial learning for generating negative examples in knowledge graph embedding algorithms. Compared with this work, both our method and KBGAN use the adversarial learning framework, but our work is fundamentally different from KBGAN. More specifically, the goal of KBGAN is to use adversarial learning to generate effective negative samples for knowledge graph embedding, while our work studies a very different problem, that is, knowledge graph alignment. We use the adversarial learning framework to learn effective mapping functions to align entities and relations across different knowledge graphs. Moreover, a major challenge in knowledge graph alignment is that the entities (or relations) in the source knowledge graph could be mapped to only a subset of entities (or relations) in the target knowledge graph. We propose to maximize the mutual information between the source entities (relations) and the mapped entities (relations) to mitigate this problem.

## 3 PROBLEM DEFINITION

**Definition 1** (KNOWLEDGE GRAPH.) *A **knowledge graph** is denoted as $\mathbf{G} = (\mathbf{E}, \mathbf{R}, \mathbf{X})$, where $\mathbf{E}$ is a set of entities, $\mathbf{R}$ is a set of relations and $\mathbf{X}$ is a set of triplets. Each triplet $\mathbf{x} = (\mathbf{h}, \mathbf{r}, \mathbf{t})$ consists of a head entity $\mathbf{h}$, a relation $\mathbf{r}$ and a tail entity $\mathbf{t}$, meaning $\mathbf{h}$ has relation $\mathbf{r}$ with $\mathbf{t}$.*

In practice, the coverage of each individual knowledge graph is usually limited, since it is typically constructed from a single source or language. To construct knowledge graphs with broader coverage, a straightforward way is to integrate multiple knowledge graphs from different sources or languages. However, each knowledge graph uses a unique symbol system to represent entities and relations, which is not compatible with other knowledge graphs. Therefore, a prerequisite for knowledge graph integration is to align entities and relations across different knowledge graphs (a.k.a., knowledge graph alignment). In this paper, we study how to align entities and relations from a source knowledge graph to those in a target knowledge graph, and the problem is formally defined below:

**Definition 2** (KNOWLEDGE GRAPH ALIGNMENT.) *Given a source knowledge graph $\mathbf{G}_s = (\mathbf{E}_s, \mathbf{R}_s, \mathbf{X}_s)$ and a target knowledge graph $\mathbf{G}_t = (\mathbf{E}_t, \mathbf{R}_t, \mathbf{X}_t)$, we aim at learning an entity alignment function $p_e$ and a relation alignment function $p_r$. Given an entity $\mathbf{e}_s$ in the source graph and an entity $\mathbf{e}_t$ in the target graph, $p_e(\mathbf{e}_t|\mathbf{e}_s)$ gives the probability that $\mathbf{e}_s$ aligns to $\mathbf{e}_t$. Similarly, for a source relation $\mathbf{r}_s$ and a target relation $\mathbf{r}_t$, $p_r(\mathbf{r}_t|\mathbf{r}_s)$ gives the probability that $\mathbf{r}_s$ aligns to $\mathbf{r}_t$.*

## 4 MODEL

In this paper we propose an unsupervised approach to learning the alignment functions, i.e., $p_e(\mathbf{e}_t|\mathbf{e}_s)$ and $p_r(\mathbf{r}_t|\mathbf{r}_s)$, for knowledge graph alignment. To learn them without supervision, we notice that we can align each source-graph triplet with a target-graph triplet by aligning the head/tail entities and relation respectively. For an ideal alignment model, all the aligned triplets should be valid ones (i.e., triplets expressing true facts). Therefore, we can improve the alignment functions by raising the plausibility of the aligned triplets. With the intuition, our approach trains a triplet discriminator to distinguish between valid and invalid triplets. Then we build a reward function from the discriminator to facilitate the alignment functions.

However, adversarial training may cause the problem of mode collapse, i.e., many entities in the source graph are aligned to only a few entities in the target graph. We avoid the problem by maximizing the mutual information between the source-graph and the aligned entities, which can effectively enforce different source-graph entities to be aligned to different target-graph entities.

The above strategies yield an unsupervised approach. However, in many cases, the structures of the source and target knowledge graphs (e.g., entity and triplet distributions) can be very different, making our unsupervised approach unable to perform effective alignment. In such cases, we can integrate our approach with existing supervised methods, and use a few labeled data as guidance, yielding a weakly-supervised approach.

### 4.1 FORMULATION OF THE ALIGNMENT FUNCTIONS

In this section, we introduce how we formulate the alignment functions, i.e., $p_e(\mathbf{e}_t|\mathbf{e}_s)$ and $p_r(\mathbf{r}_t|\mathbf{r}_s)$.

To build the alignment functions, our approach first pre-trains the entity and relation embeddings with existing knowledge graph embedding techniques (Bordes et al., 2013; Wang et al., 2014; Yang et al., 2014), where the embeddings are denoted as $\{\mathbf{v}_{\mathbf{e}_s}\}_{\mathbf{e}_s \in \mathbf{E}_s}$, $\{\mathbf{v}_{\mathbf{e}_t}\}_{\mathbf{e}_t \in \mathbf{E}_t}$ and $\{\mathbf{v}_{\mathbf{r}_s}\}_{\mathbf{r}_s \in \mathbf{R}_s}$, $\{\mathbf{v}_{\mathbf{r}_t}\}_{\mathbf{r}_t \in \mathbf{R}_t}$. In practice, our approach is flexible with any knowledge graph embedding algorithms, and we analyze some of them in Section 5.2.

The learned embeddings preserve the semantic correlations of entities and relations, thus we treat them as features and build our alignment functions on top of them. Specifically, we define the probability that a source entity $\mathbf{e}_s$ or relation $\mathbf{r}_s$ aligns to a target entity $\mathbf{e}_t$ or relation $\mathbf{r}_t$ as follows:

$$p_\theta(\mathbf{e}_t|\mathbf{e}_s) \propto \exp(-\eta||\theta_e \mathbf{v}_{\mathbf{e}_s} - \mathbf{v}_{\mathbf{e}_t}||_2^2) \quad p_\theta(\mathbf{r}_t|\mathbf{r}_s) \propto \exp(-\eta||\theta_r \mathbf{v}_{\mathbf{r}_s} - \mathbf{v}_{\mathbf{r}_t}||_2^2). \tag{1}$$

Here, $\eta$ is a temperature parameter, $\theta_e$ and $\theta_r$ are linear projection matrices, which map an entity/relation embedding in the source knowledge graph (e.g., $\mathbf{v}_{\mathbf{e}_s}$) to one in the target graph (e.g., $\theta_e \mathbf{v}_{\mathbf{e}_s}$), so that we can perform alignment by calculating the Euclidean distance between those embeddings (e.g., $\mathbf{v}_{\mathbf{e}_t}$ and $\theta_e \mathbf{v}_{\mathbf{e}_s}$).

With the definition of entity and relation alignment functions, we can further align a source-graph triplet to a target-graph triplet by aligning the head/tail entities and the relation respectively. Based on that, the probability of aligning a source-graph triplet $\mathbf{x}_s = (\mathbf{h}_s, \mathbf{r}_s, \mathbf{t}_s)$ to a target-graph triplet $\mathbf{x}_t = (\mathbf{h}_t, \mathbf{r}_t, \mathbf{t}_t)$ is given as follows:

$$p_\theta(\mathbf{x}_t|\mathbf{x}_s) = p_\theta(\mathbf{h}_t|\mathbf{h}_s)p_\theta(\mathbf{r}_t|\mathbf{r}_s)p_\theta(\mathbf{t}_t|\mathbf{t}_s). \tag{2}$$

Basically, we align the head/tail entities and the relation independently, and use the product of those probabilities to define the triplet alignment function.

By applying the triplet alignment function to all the triplets in the source graph, we obtain a distribution of the aligned triplet, which is given below:

$$p_\theta(\mathbf{x}_t) = \sum_{\mathbf{x}_s} p_d(\mathbf{x}_s)p_\theta(\mathbf{x}_t|\mathbf{x}_s) = \mathbb{E}_{p_d(\mathbf{x}_s)}[p_\theta(\mathbf{x}_t|\mathbf{x}_s)], \tag{3}$$

where $p_d(\mathbf{x}_s)$ is the data distribution of the triplets in the source graph.

### 4.2 THE ADVERSARIAL TRAINING FRAMEWORK

With the above formulation, we have obtained $p_\theta(\mathbf{x}_t)$, which is the distribution of the triplets aligned from the source graph. Intuitively, we expect every triplet sampled from the distribution to be valid ones. For this purpose, we introduce a discriminator to discriminate between valid and invalid triplets. Such a discriminator essentially estimates the plausibility of a triplet, from which we can build a reward function to guide the alignment functions.

Formally, given a triplet $\mathbf{x}_t = (\mathbf{h}_t, \mathbf{r}_t, \mathbf{t}_t)$ in the target domain, the discriminator $D_\phi$ is defined as:

$$D_\phi(\mathbf{x}_t) = \sigma(f_\phi(\mathbf{v}_{\mathbf{h}_t}) + f_\phi(\mathbf{v}_{\mathbf{t}_t}) + g_\phi(\mathbf{v}_{\mathbf{h}_t}, \mathbf{v}_{\mathbf{r}_t}, \mathbf{v}_{\mathbf{t}_t})). \tag{4}$$

Here, $\sigma$ is the sigmoid function. $f_\phi$ and $g_\phi$ are potential functions parameterized by multi-layer neural networks. The potential functions take the entity and relation embedding as input, and output a unary and a ternary potential scores to calculate $D_\phi(\mathbf{x}_t)$, which measures the probability that $\mathbf{x}_t$ is a valid triplet.

We train the discriminator $D_\phi$ by using the following objective as in Goodfellow et al. (2014):

$$O_\phi = \mathbb{E}_{p_d(\mathbf{x}_t)}[\log D_\phi(\mathbf{x}_t)] + \mathbb{E}_{p_\theta(\mathbf{x}_t)}[\log(1 - D_\phi(\mathbf{x}_t))], \tag{5}$$

where $p_d(\mathbf{x}_t)$ is the distribution of the real triplet in the target knowledge graph, and $p_\theta(\mathbf{x}_t)$ is the distribution of triplets generated by our alignment functions. Basically, the real triplets in the target knowledge graph are treated as positive examples, and those generated by our aligned functions serve as negative examples.

Based on the discriminator, we can construct a scalar-to-scalar reward function $R$ to measure the plausibility of a triplet. Then the alignment functions can be trained by maximizing the reward, and the objective function is given below:

$$O_\theta = \mathbb{E}_{p_\theta(\mathbf{x}_t)}[R(D_\phi(\mathbf{x}_t))]. \tag{6}$$

There are several ways to define the reward function $R$, which yields different adversarial training frameworks. For example, Goodfellow et al. (2014) and Ho & Ermon (2016) treat $R(x) = \log x$ as the reward function. Finn et al. (2016) uses $R(x) = \log \frac{x}{1-x}$. Che et al. (2017) considers $R(x) = \frac{x}{1-x}$. Besides, we may also leverage $R(x) = x$, which is the first-order Taylor's approximation of $-\log(1-x)$ at $x = 1$. All different reward functions essentially seek to minimize certain divergences between the data distribution $p_d(\mathbf{x}_t)$ and the model distribution $p_\theta(\mathbf{x}_t)$, and therefore they yield the same optimal solution (i.e., $p_\theta(\mathbf{x}_t) = p_d(\mathbf{x}_t)$). In practice, these reward functions may have different variance, and we empirically compare them in the experiments (Table 4).

During optimization, the derivative with respect to the alignment functions cannot be calculated directly, as the triplets sampled from the alignment functions are discrete. Therefore, we leverage the REINFORCE algorithm (Williams, 1992), which calculates the gradient as follows:

$$\nabla_\theta O_\theta = \mathbb{E}_{p_\theta(\mathbf{x}_t)}[R(D_\phi(\mathbf{x}_t))\nabla_\theta \log p_\theta(\mathbf{x}_t)]. \tag{7}$$

During training, we will alternate between optimizing the discriminator and optimizing the alignment functions, so that the discriminator can consistently provide effective supervision to benefit the alignment functions.

### 4.3 Dealing with Mode Collapse

Although the above framework provides an effective way to learn alignment functions in an unsupervised manner, the training procedure may suffer from the problem of mode collapse. More specifically, the entities in the source graph may be aligned to only a few entities in the target graph.

To avoid the problem, a natural way is to maximize the mean KL divergence between the alignment distributions of two source-graph entities $\mathbb{E}_{\mathbf{e}_{s,u},\mathbf{e}_{s,v}\sim p_d(\mathbf{e}_s)}[\text{KL}(p_\theta(\mathbf{e}_t|\mathbf{e}_{s,u}), p_\theta(\mathbf{e}_t|\mathbf{e}_{s,v}))]$. In this way, we can encourage the entities in the source graph to be aligned to different target-graph entities.

However, directly maximizing the mean divergence can be problematic. This is because the gradient of the alignment functions may explode when the mass of $p_\theta(\mathbf{e}_t|\mathbf{e}_{s,u})$ and $p_\theta(\mathbf{e}_t|\mathbf{e}_{s,v})$ concentrates in different areas (i.e., their KL divergence is very large). Due to the problem, we instead maximize a lower bound of the mean KL divergence, and a natural choice is the mutual information between the aligned entities and source-graph entities as shown in the following theorem.

**Theorem 1** *The mutual information $I(\mathbf{e}_s, \mathbf{e}_t) = \mathbb{E}_{p_\theta(\mathbf{e}_s,\mathbf{e}_t)}[\log \frac{p_\theta(\mathbf{e}_s,\mathbf{e}_t)}{p_d(\mathbf{e}_s)p_\theta(\mathbf{e}_t)}]$ provides a lower bound of the mean KL divergence between the alignment distributions of two source-graph entities $\mathbb{E}_{\mathbf{e}_{s,u},\mathbf{e}_{s,v}\sim p_d(\mathbf{e}_s)}[KL(p_\theta(\mathbf{e}_t|\mathbf{e}_{s,u}), p_\theta(\mathbf{e}_t|\mathbf{e}_{s,v}))]$.*

We prove the theorem in the appendix. With the theorem, we see that by maximizing the mutual information between the aligned entities and source-graph entities, we can guarantee the mean KL divergence not to be so small, and therefore mitigate mode collapse.

Following recent studies on neural mutual information estimation (Belghazi et al., 2018), we calculate the mutual information by introducing a function $T_\gamma$ as follows:

$$I(\mathbf{e}_s, \mathbf{e}_t) \geq I_\gamma(\mathbf{e}_s, \mathbf{e}_t) = \sup_{T_\gamma \in \mathbb{F}} \left\{ \mathbb{E}_{p_\theta(\mathbf{e}_s,\mathbf{e}_t)}[T_\gamma(\mathbf{e}_s, \mathbf{e}_t)] - \log(\mathbb{E}_{p_d(\mathbf{e}_s)p_\theta(\mathbf{e}_t)}[e^{T_\gamma(\mathbf{e}_s,\mathbf{e}_t)}]) \right\}. \tag{8}$$

Basically, $I_\gamma(\mathbf{e}_s, \mathbf{e}_t)$ is an estimation of $I(\mathbf{e}_s, \mathbf{e}_t)$, where we parameterize $T_\gamma(\mathbf{e}_s, \mathbf{e}_t)$ as a neural network, which takes the embeddings of $\mathbf{e}_s$ and $\mathbf{e}_t$ as input to output a scalar value. As we optimize $T_\gamma$, the above neural estimation will become more precise.

In most existing studies (Hjelm et al., 2018; Veličković et al., 2018), only the function $T_\gamma$ is optimized, since their end-goal is to improve representation learning by approximating the mutual information. In contrast, our end-goal is to improve the alignment function $p_\theta$ by maximizing the mutual

information $I(\mathbf{e}_s, \mathbf{e}_t)$. Therefore, besides optimizing $T_\gamma$ to tighten the bound, we also optimize $p_\theta$ to push the bound up. Specifically, the gradient for $\theta$ can be calculated as follows:

$$\nabla_\theta I_\gamma = \mathbb{E}_{p_\theta(\mathbf{e}_s, \mathbf{e}_t)}[T_\gamma \nabla_\theta \log p_\theta(\mathbf{e}_s, \mathbf{e}_t)] - \frac{\mathbb{E}_{p_d(\mathbf{e}_s)p_\theta(\mathbf{e}_t)}[e^{T_\gamma} \nabla_\theta \log p_\theta(\mathbf{e}_t)]}{\mathbb{E}_{p_d(\mathbf{e}_s)p_\theta(\mathbf{e}_t)}[e^{T_\gamma}]}, \quad (9)$$

where we again leverage the REINFORCE algorithm (Williams, 1992) to for gradient calculation. In practice, the gradient can be approximated as follows:

$$\nabla_\theta I_\gamma \simeq \sum_{i=1}^n \frac{T_\gamma(\hat{\mathbf{e}}_s^{(i)}, \hat{\mathbf{e}}_t^{(i)})\nabla_\theta \log p_\theta(\hat{\mathbf{e}}_t^{(i)}|\hat{\mathbf{e}}_s^{(i)})}{n} - \frac{\sum_{i=1}^n e^{T_\gamma(\hat{\mathbf{e}}_s^{(n+i)}, \hat{\mathbf{e}}_t^{(i)})}\nabla_\theta \log p_\theta(\hat{\mathbf{e}}_t^{(i)}|\hat{\mathbf{e}}_s^{(i)})}{\sum_{i=1}^n e^{T_\gamma(\hat{\mathbf{e}}_s^{(n+i)}, \hat{\mathbf{e}}_t^{(i)})}}, \quad (10)$$

where we have $\hat{\mathbf{e}}_s^{(i)} \sim p_d(\mathbf{e}_s)$ for $i \in [1, 2n]$, and $\hat{\mathbf{e}}_t^{(i)} \sim p_\theta(\hat{\mathbf{e}}_t^{(i)}|\hat{\mathbf{e}}_s^{(i)})$ for $i \in [1, n]$.

### 4.4 WEAKLY-SUPERVISED LEARNING

The above sections introduce an unsupervised approach to knowledge graph alignment. In many cases, the source and target knowledge graphs may have very different structures (e.g., entity or triplet distributions), making our approach fail to perform effective alignment. In these cases, we can integrate our approach with a supervised method, and leverage a few labeled data (e.g., aligned entity or relation pairs) as guidance, which yields a weakly-supervised approach.

### 4.5 OPTIMIZATION

We leverage the stochastic gradient descent algorithm for optimization. In practice, we find that first pre-training the alignment functions with existing supervised approaches, then fine-tuning them with the triplet discriminator and the mutual information maximization strategy leads to impressive results. Consequently, we adopt this framework. The detailed algorithm is presented in Algorithm 1.

## 5 EXPERIMENT

### 5.1 EXPERIMENT SETUP

Following existing studies (Zhu et al., 2017a; Chen et al., 2017a; Sun et al., 2018a), we perform evaluation on the task of entity alignment. Three different settings are considered, including supervised, weakly-supervised and unsupervised settings. Hit ratio and mean rank (MR) are reported.

Table 1: Statistics of the Datasets.

| Dataset | FB15k-1 | | FB15k-2 | | WK15k(en-fr) | | WK15k(en-de) | |
|---|---|---|---|---|---|---|---|---|
| | src | tgt | src | tgt | en | fr | en | de |
| #Entities | 14,951 | 14,951 | 14,951 | 14,951 | 15,169 | 15,392 | 15,125 | 14,602 |
| #Relations | 1,345 | 1,345 | 1,345 | 1,345 | 2,217 | 2,416 | 1,833 | 594 |
| #Triplets | 444,159 | 444,160 | 325,717 | 325,717 | 203,226 | 170,441 | 210,611 | 145,567 |
| #Training Pairs | 5,000 | | 500 | | 3,874 (en→fr) | 3,856 (fr→en) | 7,853 (en→de) | 5,606 (de→en) |
| #Test Pairs | 9,951 | | 14,451 | | 2,496 (en→fr) | 2,550 (fr→en) | 1,283 (en→de) | 1,139 (de→en) |

**1. Datasets.** We use four datasets in experiment, and their statistics are available in Table 1.

- **FB15k-1, FB15k-2**: Following Zhu et al. (2017a), we construct two datasets from the FB15k dataset (Bordes et al., 2013). In FB15k-1, the two knowledge graphs share 50% triplets, and in FB15k-2 10% triplets are shared. According to the study, we use 5000 and 500 aligned entity pairs as labeled data in FB15k-1 and FB15k-2 respectively, and the rest for evaluation.
- **WK15k(en-fr)**: A bi-lingual (English and French) dataset in Chen et al. (2017a). Some aligned triplets are provided as labeled data, and some aligned entity pairs as test data. The labeled data and test data have some overlaps, so we delete the overlapped pairs from labeled data. Also, some entities in the test set are not included in the training set, and thus we filter out those entities.
- **WK15k(en-de)**: A bi-lingual (English and German) dataset used in Chen et al. (2017a). The dataset is similar to WK15k(en-fr), so we perform preprocessing in the same way.

**2. Compared Algorithms.** (1) **iTransE** (Zhu et al., 2017a): A supervised method for knowledge graph alignment. (2) **MLKGA** (Chen et al., 2017a): A supervised method for multi-lingual

knowledge graph alignment. (3) **AlignE** (Sun et al., 2018a): A supervised method for knowledge graph alignment, which leverages a bootstrapping manner for training. (4) **BootEA** (Sun et al., 2018a): Another bootstrapping method for knowledge graph alignment. (5) **Procrustes** (Artetxe et al., 2017): A supervised method for word translation, which learns the translation in a bootstrapping way. We apply the method on the pre-trained entity and relation embeddings to perform knowledge graph alignment. (6) **UWT** (Conneau et al., 2017): An unsupervised word translation method, which leverages adversarial training and a refinement strategy. We apply the method to the entity and relation embeddings to perform alignment. (7) **KAGAN**: Our proposed approach, which uses both the triplet discriminator and the mutual information maximization strategy for training.

**3. Parameter Settings.** For all datasets, 10% labeled pairs are treated as the validation set, which is used for hyper-parameter selection for each compared algorithm. For the dimension of the entity embedding, we choose the optimal value from $\{64, 128, 256, 512\}$ based on the performance on the validation set. For our proposed approach, the entity and relation embeddings are trained with the TransE (Bordes et al., 2013) algorithm by default, because of its simplicity and effectiveness. The alignment functions are pre-trained with the Procrustes (Artetxe et al., 2017) algorithm in the weakly-supervised and supervised settings, because Procrustes is both effective and efficient. For the potential functions $f_\phi$ and $g_\phi$ in the discriminator, and the T function $T_\gamma$ in the neural estimator of mutual information, we build each of them using a two-layer neural network with 2048 hidden units and the LeakyReLU activation function (Maas et al.). SGD is used for optimization. The learning rates for the triplet discriminator and the mutual information estimator are set as 0.1 during pre-training, and 0.001 during training. The learning rate for the alignment functions is set as 0.001. Early stopping is used during training.

## 5.2 EXPERIMENT RESULTS

Table 2: Results of Entity Alignment on the WK datasets.

| Setting | Algorithm | WK15k fr2en | | | WK15k en2fr | | | WK15k de2en | | | WK15k en2de | | |
|---|---|---|---|---|---|---|---|---|---|---|---|---|---|
| | | H@1 | H@10 | MR | H@1 | H@10 | MR | H@1 | H@10 | MR | H@1 | H@10 | MR |
| Unsupervised | UWT | 0.66 | 2.97 | 6099.0 | 0.03 | 0.46 | 6091.0 | 0.44 | 1.55 | 5910.3 | 0.55 | 3.06 | 2982.5 |
| | KAGAN | **1.22** | **4.59** | **5798.9** | **0.24** | **1.32** | **5696.0** | **0.61** | **2.37** | **2939.2** | **0.78** | **4.99** | **2134.9** |
| Supervised | iTransE | 0.94 | 12.59 | 3192.1 | 0.64 | 13.94 | 2922.3 | 5.36 | 12.55 | 4048.2 | 8.11 | 16.13 | 1803.2 |
| | MLKGA | 26.63 | 62.43 | 176.0 | 26.20 | 62.74 | 193.6 | 60.40 | 81.30 | 93.2 | 46.92 | 72.80 | 113.9 |
| | AlignE | 15.29 | 46.12 | 523.0 | 9.98 | 37.98 | 429.1 | 26.08 | 43.63 | 300.0 | 19.02 | 40.14 | 408.2 |
| | BootEA | 32.30 | 60.59 | 392.9 | 31.45 | 56.97 | 317.2 | 41.00 | 58.74 | 195.9 | 35.23 | 55.73 | 334.8 |
| | Procrustes | 32.24 | 67.37 | 139.3 | 30.97 | 64.58 | 173.8 | 64.44 | 83.76 | 89.0 | 48.17 | 73.97 | 113.8 |
| | KAGAN | **35.88** | **68.59** | **136.3** | **35.54** | **68.23** | **165.4** | **67.55** | **85.07** | **68.9** | **51.13** | **74.43** | **106.9** |

Table 3: Results of Entity Alignment on the FB datasets.

| Setting | Algorithm | FB15k-1 | | | FB15k-2 | | |
|---|---|---|---|---|---|---|---|
| | | H@1 | H@10 | MR | H@1 | H@10 | MR |
| Unsupervised | UWT | 79.33 | 91.48 | 18.6 | 70.03 | 86.86 | 29.6 |
| | KAGAN | **83.41** | **92.63** | **10.5** | **73.68** | **88.91** | **26.3** |
| Supervised | iTransE | 64.58 | 80.87 | 47.0 | 9.69 | 29.23 | 760.7 |
| | MLKGA | 78.87 | 90.66 | 24.3 | 53.60 | 78.80 | 66.6 |
| | AlignE | 57.94 | 77.51 | 63.9 | 17.76 | 43.40 | 223.0 |
| | BootEA | 74.98 | 88.25 | 21.8 | 20.05 | 46.29 | 216.6 |
| | Procrustes | 82.36 | 92.13 | 15.4 | 72.08 | 87.15 | 28.3 |
| | KAGAN | **84.76** | **93.68** | **9.9** | **73.73** | **88.80** | **24.8** |

Table 4: Study of Reward Functions.

| Method | WK15k fr2en | | |
|---|---|---|---|
| | H@1 | H@10 | MR |
| w/o reward | 32.24 | 67.37 | 139.3 |
| $\log x$ | 35.25 | 67.10 | 149.2 |
| $\log \frac{x}{1-x}$ | 35.37 | 67.76 | 148.9 |
| $\frac{x}{1-x}$ | **36.00** | 68.27 | 138.9 |
| $x$ | 35.88 | **68.59** | **136.3** |

(a) WK15k fr2en     (b) WK15k en2fr     (c) WK15k de2en     (d) WK15k en2de

Figure 1: Performance in the weakly-supervised setting.

**1. Comparison with Baseline Methods.** The main results are presented in Table 2 and 3. In the supervised setting, our approach significantly outperforms all the compared methods, showing our

approach can utilize the labeled data more effectively. In the unsupervised setting on FB15k datasets, without using any labeled data, our approach already achieves close results as in supervised settings.

However, the performance on WK15k in the unsupervised setting is quite poor. The reason is that the source and target knowledge graphs in WK15k have very different structures (i.e., entity distribution and triplet distribution). Therefore, the triplet discriminator cannot well discriminate between the real and fake triplets, and further provides effective reward. In such cases, we may leverage a few aligned entity pairs to pre-train our alignment functions, leading to a weakly-supervised approach. We present the results of this weakly-supervised approach in Figure 1. The Procrustes algorithm (Artetxe et al., 2017) is chosen as the compared method, since it has the best performance in the supervised setting. From the results, we see that by using a very small number of aligned pairs, our approach (blue line) already outperforms Procrustes in the supervised setting (black line), showing that our approach is also quite effective in the weakly-supervised setting.

2. **Analysis of Mutual Information Maximization.** In KAGAN, we avoid mode collapse by maximizing the mutual information between the source-graph entities and the aligned entities. To understand its effect, we conduct some ablation studies in the supervised setting. Table 5 presents the results. With mutual information maximization, we consistently get better results, which proves the effectiveness of such a strategy. We also conduct some cases studies in appendix (see Section B).

Table 5: Analysis of Mutual Information Maximization.

| Method | FB15k-1 | | | WK15k de2en | | | WK15k en2de | | |
|---|---|---|---|---|---|---|---|---|---|
| | H@1 | H@10 | MR | H@1 | H@10 | MR | H@1 | H@10 | MR |
| w/o MI | 83.84 | 92.60 | 11.5 | 66.55 | 84.72 | 83.0 | 50.43 | 74.12 | 111.7 |
| with MI | **84.76** | **93.68** | **9.9** | **67.55** | **85.07** | **68.9** | **51.13** | **74.43** | **106.9** |

Table 6: Analysis of the Discriminator Training.

| Method | FB15k-2 | | | WK15k fr2en | | |
|---|---|---|---|---|---|---|
| | H@1 | H@10 | MR | H@1 | H@10 | MR |
| Rand. | 68.72 | 81.34 | 37.8 | 32.90 | 67.22 | 178.9 |
| Rand.+Adv. | 72.72 | 88.34 | 28.0 | 33.88 | 66.86 | 166.2 |
| Adv. | **73.68** | **88.91** | **26.3** | **35.88** | **68.59** | **136.3** |

Table 7: Comparison of Embedding Methods.

| Method | FB15k-2 | | | WK15k fr2en | | |
|---|---|---|---|---|---|---|
| | H@1 | H@10 | MR | H@1 | H@10 | MR |
| TransE | **73.68** | **88.91** | **26.3** | **35.88** | **68.59** | **136.3** |
| TransH | 34.39 | 47.99 | 464.2 | 12.04 | 25.41 | 1493.6 |
| DistMult | 0.15 | 0.31 | 5351.2 | 0.12 | 0.20 | 5754.8 |

3. **Analysis of the Discriminator Training.** In our approach, a discriminator is trained to discriminate between the real and fake triplets. During discriminator training, we choose the triplets generated by our alignment models as fake triplets by default, and there are also some other ways to generate the fake triplets. In this section, we compare different options of the fake triplets. Our default method, which treats the generated triplets as fake ones, is denoted as "Adv.". Another common choice is to use random triplets as fake ones, as used in most knowledge graph embedding algorithms. We denote this variant as "Rand.". Besides, we can also leverage both the random and the generated triplets as fake ones, and such a method is denoted as "Rand.+Adv.".

We compare the three variants on the FB15k-2 dataset (unsupervised setting) and the WK15k datasets (supervised setting), and the results are presented in Table 6. We see that using random triplets as fake ones ("rand." and "rand.+adv.") leading to inferior results compared with using only generated triplets, which proves the effectiveness of our adversarial training framework.

4. **Comparison of Knowledge Graph Embeddings.** In our approach, we pre-train the entity and relation embeddings with existing knowledge graph embedding algorithms, and then use these embeddings as features for training the alignment functions. Our approach is compatible with a wide range of knowledge graph emebedding algorithms. In this section, we compare different knowledge graph embedding algorithms. We choose three commonly-used embedding algorithms, including TransE (Bordes et al., 2013), TransH (Wang et al., 2014) and DistMult (Yang et al., 2014).

The results on the FB15k-2 dataset (unsupervised setting) and the WK15k dataset (supervised setting) are presented in Table 7. We see that TransE achieves the best performance among all three algorithms. The reason is that TransE uses a linear scoring function, and the relations are characterized as linear translations in the embedding space. The information encoded in the learned embeddings can be effectively recovered by a linear alignment function, as used in our approach. In contrast, TransH and DistMult use more complicated scoring functions, and the information in the learned embeddings cannot be well recovered by a simple linear alignment function. In the future, we plan to explore some nonlinear alignment functions to further improve the performance.

**5. Comparison of Reward Functions.** In our approach, we can choose different reward functions, leading to different adversarial training frameworks. These frameworks have the same optimal solution, but with different variance. Next, we compare them on WK15k in the supervised setting, and the results are presented in Table 4. We notice that all reward functions lead to significant improvement compared with using no reward. Among them, $\frac{x}{1-x}$ and $x$ obtain relatively better results.

## 6 CONCLUSION

This paper studies knowledge graph alignment, and an unsupervised approach is proposed based on adversarial training and mutual information maximization, which can also be seamlessly integrated with existing supervised methods for weakly-supervised learning. Experimental results on several real datasets prove the effectiveness of our approach in both the unsupervised and weakly-supervised settings. In the future, we plan to learn alignment functions from two directions (source to target and target to source) to further improve the results, which is similar to CycleGAN (Zhu et al., 2017b).

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

## A    PROOF OF THE THEOREM 1

We first restate the theorem and then give the proof.

**Theorem 2** *The mutual information $I(\mathbf{e}_s, \mathbf{e}_t) = \mathbb{E}_{p_\theta(\mathbf{e}_s, \mathbf{e}_t)}[\log \frac{p_\theta(\mathbf{e}_s, \mathbf{e}_t)}{p_d(\mathbf{e}_s)p_\theta(\mathbf{e}_t)}]$ provides a lower bound of the mean KL divergence between the alignment distributions of two source-graph entities $\mathbb{E}_{\mathbf{e}_{s,u}, \mathbf{e}_{s,v} \sim p_d(\mathbf{e}_s)}[KL(p_\theta(\mathbf{e}_t|\mathbf{e}_{s,u}), p_\theta(\mathbf{e}_t|\mathbf{e}_{s,v})))]$.*

**Proof 1** *For the mean KL divergence, we have:*

$$\mathbb{E}_{\mathbf{e}_{s,u}, \mathbf{e}_{s,v} \sim p_d(\mathbf{e}_s)}[KL(p_\theta(\mathbf{e}_t|\mathbf{e}_{s,u})||p_\theta(\mathbf{e}_t|\mathbf{e}_{s,v})))] =$$

$$\mathbb{E}_{p_d(\mathbf{e}_{s,u})p_d(\mathbf{e}_{s,v})p_\theta(\mathbf{e}_t|\mathbf{e}_{s,u})}[\log p_\theta(\mathbf{e}_t|\mathbf{e}_{s,u})] - \mathbb{E}_{p_d(\mathbf{e}_{s,u})p_d(\mathbf{e}_{s,v})p_\theta(\mathbf{e}_t|\mathbf{e}_{s,u})}[\log p_\theta(\mathbf{e}_t|\mathbf{e}_{s,v})]$$

*For the first term, it equals to $\mathbb{E}_{p_\theta(\mathbf{e}_t, \mathbf{e}_{s,u})}[\log p_\theta(\mathbf{e}_t|\mathbf{e}_{s,u})]$. For the second term, we have:*

$$\mathbb{E}_{p_d(\mathbf{e}_{s,u})p_d(\mathbf{e}_{s,v})p_\theta(\mathbf{e}_t|\mathbf{e}_{s,u})}[\log p_\theta(\mathbf{e}_t|\mathbf{e}_{s,v})] = \mathbb{E}_{p_\theta(\mathbf{e}_t, \mathbf{e}_{s,u})}[\mathbb{E}_{p_d(\mathbf{e}_{s,v})}[\log p_\theta(\mathbf{e}_t|\mathbf{e}_{s,v})]]$$

$$\leq \mathbb{E}_{p_\theta(\mathbf{e}_t, \mathbf{e}_{s,u})}[\log \mathbb{E}_{p_d(\mathbf{e}_{s,v})}[p_\theta(\mathbf{e}_t|\mathbf{e}_{s,v})]] = E_{p_\theta(\mathbf{e}_t, \mathbf{e}_{s,u})}[\log p_\theta(\mathbf{e}_t, \mathbf{e}_{s,v})]$$

*Here, the inequation is based on the Jensen's inequality ($\log \mathbb{E}[f(x)] \geq \mathbb{E}[\log f(x)]$). By combing the above terms, we obtain:*

$$\mathbb{E}_{\mathbf{e}_{s,u}, \mathbf{e}_{s,v} \sim p_d(\mathbf{e}_s)}[KL(p_\theta(\mathbf{e}_t|\mathbf{e}_{s,u})||p_\theta(\mathbf{e}_t|\mathbf{e}_{s,v})))]$$

$$= \mathbb{E}_{p_\theta(\mathbf{e}_t, \mathbf{e}_{s,u})}[\log p_\theta(\mathbf{e}_t|\mathbf{e}_{s,u})] - E_{p_\theta(\mathbf{e}_t, \mathbf{e}_{s,u})}[\log p_\theta(\mathbf{e}_t, \mathbf{e}_{s,v})]$$

$$\geq \mathbb{E}_{p_d(\mathbf{e}_s)}[KL(p_\theta(\mathbf{e}_t|\mathbf{e}_s)||p_\theta(\mathbf{e}_t)))] = I(\mathbf{e}_s, \mathbf{e}_t)$$

*The theorem is proved.*

## B    MORE ANALYSIS ON MUTUAL INFORMATION MAXIMIZATION

To further understand the effect of mutual information maximization, we show some case study results on the WK15k datasets in Fig. 2. For each entity in the target knowledge graph, we count how many source-graph entities are aligned to that entity. Then we find top 100 target-graph entities with the largest counts, and their counts are reported. From the figure, we see that by maximizing the mutual information, the alignment counts of the top-ranked entities become smaller, which proves that our method can indeed encourage different source-graph entities to be aligned to different target-graph entities, and thus alleviate mode collapse.

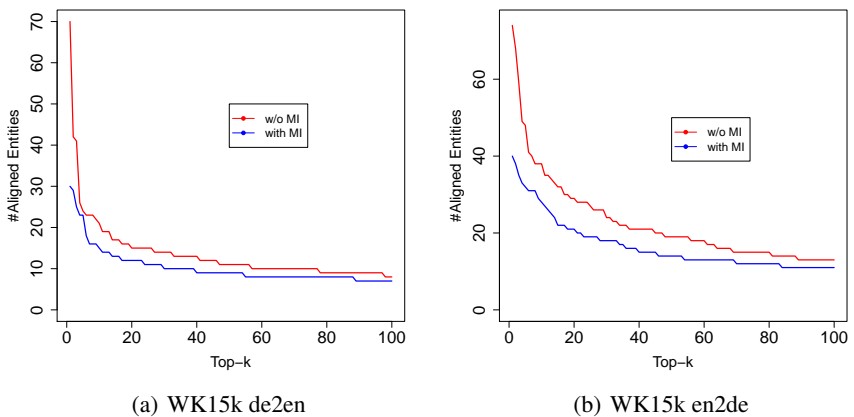

(a) WK15k de2en      (b) WK15k en2de

Figure 2: Case study of mutual information maximization.

## C    OPTIMIZATION ALGORITHM OF KAGAN

We present the detailed optimization algorithm of our approach as follows:

---

**Algorithm 1** Optimization Algorithm

---

1: **Input:** Two knowledge graphs $\mathbf{G}_s$ and $\mathbf{G}_s$, some aligned entity/relation pairs (optional).
2: **Output:** The alignment functions $p_\theta$.
3: Pre-train the alignment functions with the aligned pairs.
4: Pre-train the triplet discriminator $D_\phi$ according to Equation 5.
5: Pre-train the mutual information estimator $I_\gamma$ according to Equation 8.
6: **while** not converge **do**
7:     Update the triplet discriminator $D_\phi$ according to Equation 5.
8:     Update the alignment functions $p_\theta$ with $D_\phi$ according to Equation 7.
9:     Update the mutual information estimator $I_\gamma$ according to Equation 8.
10:     Update the alignment functions $p_\theta$ to maximize $I_\gamma$ according to Equation 10.
11: **end while**

---

# D    ADDITIONAL EXPERIMENTS ON LARGE DATASETS

Table 8: Statistics of WK120k(en-de).

| Dataset | WK120k(en-de) | |
|---|---|---|
| | en | de |
| #Entities | 67,648 | 61,941 |
| #Relations | 2,381 | 858 |
| #Triplets | 626,593 | 391,044 |
| #Training Pairs | 22,934 (en→de) | 18,187 (de→en) |
| #Test Pairs | 6,173 (en→de) | 4,819 (de→en) |

We also conduct some experiments on WK120k(en-de), which is a dataset in Chen et al. (2017a). Similar to WK15k, as the training data and test data have some overlap, we filter out those over-lapped data in the training set. Also, some entities in the test set are not included in the training set, and thus we remove those entities in the test set. The detailed statistics are summarized in Table 8.

Table 9: Results of Entity Alignment on the WK120k datasets.

| Algorithm | WK120k de2en | | | WK120k en2de | | |
|---|---|---|---|---|---|---|
| | H@1 | H@10 | MR | H@1 | H@10 | MR |
| MLKGA | 9.42 | 26.96 | 3380.9 | 7.66 | 16.96 | 4116.5 |
| AlignE | 8.74 | 22.54 | 5877.1 | 5.17 | 15.91 | 8629.2 |
| BootEA | 16.68 | 29.88 | 4649.9 | 10.54 | 19.99 | 8002.7 |
| Procrustes | 20.73 | 38.66 | 3034.6 | 13.85 | 24.93 | 4239.4 |
| KAGAN | **23.10** | **39.95** | **2873.8** | **14.60** | **25.69** | **3664.1** |

The results of the compared methods are presented in Table 9. We see that our proposed approach outperforms all the baseline methods, which shows the effectiveness of KAGAN on large datasets.

