# OpenReview forum: "Weakly-supervised Knowledge Graph Alignment with Adversarial Learning"
_ICLR.cc/2020/Conference — Reject_

### Official Review · AnonReviewer3 · 2019-10-18
**Official Blind Review #3**

**Rating:** 6

**Review:**

The authors aim at learning alignments between knowledge graphs. For this they use a discriminator that functions as an adversary to the parameterized triple alignment function leading to adversarial training. Furthermore, they regularize the training by maximizing a neural mutual information estimator. This requires training another approximating neural network along the way.
Several experiments seem to indicate that their approach improves over competing methods in the unsupervised and weakly-supervised setting.

**Experience Assessment:**

I do not know much about this area.

**Review Assessment: Checking Correctness Of Derivations And Theory:**

I assessed the sensibility of the derivations and theory.

**Review Assessment: Checking Correctness Of Experiments:**

I assessed the sensibility of the experiments.

**Review Assessment: Thoroughness In Paper Reading:**

I read the paper at least twice and used my best judgement in assessing the paper.

---

> ### Author Response · Authors · 2019-11-13
> **Response to Reviewer #3**
>
> Thank you for your time and the comments!

---

### Official Review · AnonReviewer1 · 2019-10-24
**Official Blind Review #1**

**Rating:** 6

**Review:**

The authors propose an unsupervised knowledge graph alignment framework, KAGAN, using adversarial training. Specifically, they utilize a triple discriminator to discriminate the aligned source triples and target triples, and a reward function to minimize the divergence between source triples and target triples. Also, they propose to leverage the lower bound of the mean KL divergence (mutual information) to resolve the mode collapse problem. The proposed method can be incorporated with a supervised method to be a weakly-supervised approach. Even though there are a family of unsupervised approaches for domain alignment, this paper is the first to solve the knowledge graph alignment problem in an unsupervised/weakly supervised way.

Strength:
1.	The paper addresses a critical knowledge graph alignment problem using GAN based on triplets, not usually entity alignment in literature, but also considers the relation alignment in knowledge graph.
2.	The paper tries to solve the problem in an unsupervised way, and shows the on-par performance with weakly supervised methods in FB15k dataset.
3.	The paper considers mode collapse problem and tries to solve the problem via mutual information rather than mean KL divergence, also gives the theoretical proof.
4.	Detailed experimental analysis and ablation studies show the effectiveness of the proposed method on small datasets.
5.	The paper is well-written and easy to follow.


I have two concerns as follows:
1.	The authors conduct experiments on multiple small KG datasets such as FB15K and WK15K. But the reviewer finds that the baseline papers authors mentioned also have experiments on larger datasets like WK120K (Chen et al., 2017a), WK60k, DBP-WD or DBP YG (Sun et al., 2018a). It is essential to conduct the experiment on larger datasets to verify the effectiveness of the proposed method.
2.	Is it necessary to construct a reward function to update the alignment function using REINFORCE algorithm? For instance, current distribution matching methods can define the discrepancy between the two distributions (such as target distribution and aligned source distribution). It can directly optimize the loss in an end-to-end differentiable way instead of a reinforcement learning way. It can avoid the sampling and provide a more stable optimization process.


**Experience Assessment:**

I have published one or two papers in this area.

**Review Assessment: Checking Correctness Of Derivations And Theory:**

I assessed the sensibility of the derivations and theory.

**Review Assessment: Checking Correctness Of Experiments:**

I carefully checked the experiments.

**Review Assessment: Thoroughness In Paper Reading:**

I read the paper thoroughly.

---

> ### Author Response · Authors · 2019-11-13
> **Response to Reviewer #1**
>
> Thank you for the insightful comments!
>
> 1. It is a very good point to evaluate our proposed approach on larger datasets! Following your suggestions, we have conducted additional experiments on the WK120k dataset (Chen et al., 2017a), and the results are presented as follows:
>
> ----------------------------------------------------------------------------------
>                      |         de2en                   |         en2de
>  Algorithm |-----------------------------------------------------------------
>                     |  H@1  |  H@10 |  MR  |  H@1  |  H@10 |  MR
> ----------------------------------------------------------------------------------
> MLKGA      |  9.42 | 26.96 | 3381   |  7.66   | 16.96 | 4117
> AlignE        |  8.74 | 22.54 | 5877   |  5.17   | 15.91 | 8629
> BootEA      | 16.68 | 29.88 | 4650   | 10.54 | 19.99 | 8003
> Procrustes | 20.73 | 38.66 | 3034  | 13.85 | 24.93 | 4239
> ----------------------------------------------------------------------------------
> KAGAN      | 23.10 | 39.95 | 2874   | 14.60 | 25.69 | 3664
> ----------------------------------------------------------------------------------
>
> We can see that our proposed approach (KAGAN) outperforms all the compared methods, showing its effectiveness on larger datasets. We have added the results to the Appendix D of the revised draft.
>
> 2. It is an insightful suggestion to use distribution matching methods for knowledge graph alignment, as these methods are more stable in terms of optimization.
>
> The distribution matching methods typically match the distributions of entity embeddings and relation embeddings independently between the source and target knowledge graphs. However, they ignore matching the local knowledge graph structures (i.e., triplets) during alignment. Such local structure matching can be achieved by our approach through using the REINFORCE algorithm. More specifically, our approach aligns each source triplet to a concrete target triplet, and further treats the plausibility of the target triplet as reward to adjust the alignment function.
>
> Indeed, in our experiment, UWT (Conneau et al., 2017) is a baseline algorithm which uses distribution matching for alignments. As shown in the paper, our approach outperforms UWT in the unsupervised setting. For the supervised setting, we conduct additional experiments on the WK15k(en-fr) dataset, where we use the same supervised method for pre-training, and then train the alignment functions with either distribution matching or REINFORCE. The Hit@1 of the distribution matching method is 25.60 (en2fr) and 25.45 (fr2en), whereas the Hit@1 of our approach is 35.88 (fr2en) and 35.54 (en2fr), which indicates the effectiveness of our method.
>
> Again, thank you for the valuable comments. If you have any further concerns, we are glad to answer.

---

### Official Review · AnonReviewer2 · 2019-10-28
**Official Blind Review #2**

**Rating:** 3

**Review:**

The authors propose a formulation to align entities and relation across different knowledge-basis. Each entity (or relation) is associated with a distribution over the entities (or relations) in the other KG through an exponential kernel-density-like-estimate.


The main contribution seems to be the use of GAN to generate good negative triplets instead of the traditional search-for-hard-negatives in constrained problems. This is a novel contribution and I see the value of this line of work.

The reason why I am tending not to accept this work is because it very similar(/same) to existing work like KBGAN (https://www.aclweb.org/anthology/N18-1133/). Most of the key ideas have already been covered there, and I would like to see a comparison of this work with that before acceptance.


nits:
some column names in the table are inconsistent - hit vs h, mrr vs mr.


**Experience Assessment:**

I have read many papers in this area.

**Review Assessment: Checking Correctness Of Derivations And Theory:**

I did not assess the derivations or theory.

**Review Assessment: Checking Correctness Of Experiments:**

I assessed the sensibility of the experiments.

**Review Assessment: Thoroughness In Paper Reading:**

I read the paper at least twice and used my best judgement in assessing the paper.

---

> ### Author Response · Authors · 2019-11-10
> **About the Comparison with the KBGAN Paper**
>
> Thank you for the comments and also pointing out the related paper. We are aware of this paper. Indeed, both our paper and the KBGAN paper use the adversarial learning framework, which has been widely used for a variety of problems. However, our paper is fundamentally different from the KBGAN paper. The goal of the KBGAN paper is to use the GAN framework to generate effective negative samples for knowledge graph embedding, while our work studies a very different problem, that is, knowledge graph alignment. We use the adversarial learning framework to learn effective mapping functions to align entities and relations across different knowledge graphs. Moreover, a major challenge in knowledge graph alignment is that the entities (or relations) in the source knowledge graph could be mapped to only a subset of entities (or relations) in the target knowledge graph. We propose to maximize the mutual information between the source entities (relations) and the mapped entities (relations) to mitigate this problem.
>
> According to your comments, we have cited the KBGAN paper and discussed it in related work, and also fixed the typos you point out.
>
> Again, thank you for the valuable suggestions. If you still have any concerns or questions, we are happy to answer.

---

### Author Response · Authors · 2019-11-13
**Summary of Revision**

We would like to thank all the reviewers for the helpful comments and suggestions!
Based on the comments, we have made the following changes in the revised draft.

1. We have added the comparison with the KBGAN paper (Cai & Wang, 2018) in the related work section, based on the suggestions from the reviewer #2.

2. We have added the experiments on a larger dataset (i.e., WK120k from Chen et al., 2017a) in the appendix D, based on the suggestions from the reviewer #1.

3. We have fixed the typos pointed out by the reviewer #2.

---

### Decision · Program_Chairs · 2019-12-19

**Decision:**

Reject

**Comment:**

Thanks for your detailed feedback to the reviewers, which clarified us a lot in many respects.
This paper potentially discusses an interesting problem, and the concern raised by Review #2 was addressed in the revised paper.
However,  given the  high competition at ICLR2020, this paper is unfortunately below the bar.
We hope that the reviewers' comments are useful for improving the paper for potential future publication.

The